# Human Serum Albumin Misfolding in Aging and Disease

**DOI:** 10.3390/ijms231911675

**Published:** 2022-10-02

**Authors:** Francis H. C. Tsao, Keith C. Meyer

**Affiliations:** Department of Medicine, Division of Pulmonary and Critical Care Medicine, School of Medicine and Public Health, University of Wisconsin, Madison, WI 53792, USA

**Keywords:** age, albumin, inflammation, misfolding, serum, sPLA2

## Abstract

Age-dependent conformational stability of human serum albumin was determined by the method of fluorescent bilayer liposome assay. After pre-heating at 80 °C, albumin in the sera of 74-year-old healthy subjects exhibited hydrophobic effects on liposomes and made liposomal membrane phospholipids more susceptible to hydrolysis by the lipolytic enzyme phospholipase A2. In contrast, albumin in the sera of 24-year-old individuals was stable at 80 °C and displayed no increased hydrophobic effects on liposomes. The results suggest that albumin in the sera of 74-year-old subjects is more easily converted to a misfolded form in which its protein structure is altered when compared to albumin in the sera of 24-year-old individuals. Misfolded albumin can lose its ability to carry out its normal homeostatic functions and may promote alterations in membrane integrity under inflammatory conditions. However, our investigation has limitations that include the lack of testing sera from large numbers of individuals across a broad range of age to validate our preliminary observations of age-dependent differences in albumin stability and its interactions with liposomes.

## 1. Introduction

Albumin is the most abundant protein in circulating blood and transports a number of life-supporting molecules including free fatty acids (FA) to cells throughout the body. Produced in the liver, albumin is also the most important protein for maintaining proper oncotic pressure to keep fluids from leaking out of blood vessels. Approximately 40% of albumin is found in plasma, while 60% is found in tissue interstitium [1]. Serum albumin levels can be depressed with nutritional deficiency and medical conditions, such as liver or kidney disease. Low levels of serum albumin are closely associated with the incidence of morbidity and mortality in hospitalized patients [2]. Despite the wide use of serum albumin levels for evaluation of patient’s prognosis, little is known about the correlation between the protein albumin quality, such as structural integrity and functional activity, and a patient’s pathophysiological conditions. It is known that binding of a FA molecule to each FA-binding site of human serum albumin is accompanied by a change in the albumin conformational structure [3]. In recent years, more evidence has shown that protein conformational change or misfolding and aggregation is associated with the onset of numerous diseases [4,5]. In advanced age, proteins may be susceptible to conformational changes in their structures that can cause protein misfolding and aggregation [6,7]. The most notable aging-related protein misfolding diseases are the neurodegenerative diseases, including the intracellular α-synuclein misfolding in neurons in Parkinson’s disease and the extracellular amyloid-β aggregations as well as the intracellular neurofibrillary tangles of Tau protein in Alzheimer’s disease [8,9].

Although it is known that human serum albumin can undergo conformational changes upon ligand bindings and molecular modifications [3,10], the pathophysiological significance of albumin conformational change is not clear. In this paper, we discuss the potential role of albumin misfolding on bilayer membrane integrity in aging and disease states. We previously suggested that serum albumin might play a critical role to bind and remove FA liberated from membranes of cells injured under infection-induced systemic inflammation [11]. Under such conditions, concentrations of an inflammatory response, 14,000 molecular weight secretory phospholipase A2 (sPLA2), were increased in the serum of both humans and laboratory animals infected with bacteria [12,13]. We suggested that an important role of increased amounts of sPLA2 was to hydrolyze the externalized, negatively charged phospholipids from membranes of apoptotic cells, in combination with the removal of the released FA from membranes by albumin to maintain homeostatic membrane integrity [11]. Interestingly, elevation of serum sPLA2 was accompanied by a decrease in serum albumin interactions with cell membranes as well as decreased activity of a specific fraction of albumin (SFA) in patients with bacterial sepsis [11]. Our data suggest that saturation of albumin-FA binding sites with pre-bound FA derived from sPLA2-hydrolyzed membrane phospholipids totally depleted the SFA activity [11]. The sPLA2-SFA inverse activity relationship was further corroborated by the analysis of a large number of serum samples from patients with sepsis [14]. It is clear that the impaired serum albumin (SFA)–membrane interaction activity in patients with sepsis was due to albumin-FA binding sites being occupied by pre-bound FA, which were presumably derived from sPLA2-hydrolyzed phospholipids in membranes of massively injured cells in sepsis [14]. Pre-bound FA derived from injured cell membranes taken up by albumin not only depleted albumin-FA binding site capacity, it also caused an albumin conformational change, which increased albumin hydrophobic interactions with bilayer membranes, and consequently, enhanced the sPLA2 activity [14]. These results appear to be consistent with the dogma of binding of FA molecules to human serum albumin leading to a change in the albumin conformational structure [3].

Hydrophobic interactions between albumin and bilayer membranes are mainly governed not only by albumin’s conformational structure, but also by the degree of membrane fluidity. We previously used bilayer liposomes with a high degree of membrane fluidity as substrates for the sPLA2 catalytic reaction to determine the effect of serum albumin on sPLA2 activity [15]. We found that albumin in sera collected from healthy 74-year-old individuals enhanced sPLA2 activity much greater than sera from 24-year-old healthy subjects [15]. The markedly increased sPLA2 activity in serum from the 74-year-old subjects was due to enhanced hydrophobic interactions between albumin and liposomal membranes. These results suggest that serum albumin of the 74-year-old subjects had a conformational structure different from the albumin in serum from the 24-year-old subjects. Albumin from the older subjects’ serum likely had its hydrophobic amino acid residues exposed to the aqueous phase due to its misfolded structure. Under these assay conditions, the hydrophobic amino acid residues of misfolded albumin can exert hydrophobic effects that make membranes more accessible to the hydrolysis of phospholipids at membrane interface by sPLA2. Although differences of albumin misfolding in the sera from the healthy 74-year-old volunteers, elderly subjects with mild cognitive impairment, and elderly patients with Alzheimer’s disease could not be directly determined, misfolded albumin in the cerebrospinal fluid of patients with Alzheimer’s appeared to have greater hydrophobic effects on bilayer membranes [15]. The purpose of this study was to determine the hydrophobic effects of serum albumin on bilayer membranes based on albumin protein conformation, rather than on membrane fluidity, and to find out whether albumin structural integrity changes with age.

## 2. Results

In our previous study, aberrantly folded albumin in sera of 74-year-old subjects was indirectly determined by using high-membrane-fluidity bilayer liposomes as sPLA2 substrates to detect the protein-membrane hydrophobic effect [15]. We have also previously determined that the effects of serum albumin of 24-year-old and 74-year-old subjects on the sPLA2 activity exhibited no significant difference when using low-membrane-fluidity bilayer liposome substrates under the same assay conditions [15]. In this study, we again observed that there was no difference between the albumin in the sera of 24-year-old and in the sera of 74-year-old subjects on the effects on sPLA2 activity using low-membrane-fluidity liposome substrates after sera were pre-incubated at 22 °C for 10 min (Figure 1 Column 1 and Column 3). This effect was presented as “Albumin Activity”, which represents the fluorescence intensity change due to albumin-membrane interactions, albumin-FA binding, and sPLA2 catalytic activity in the assay. Unlike serum pre-incubated at 22 °C, the albumin activity in the serum of 74-year-old persons after pre-incubation at 80 °C for 10 min (Figure 1 Column 4) was markedly higher than that treated at 22 °C (Figure 1 Column 3) (*p* < 0.0001). The increase in albumin activity was due to the hydrophobic effects between misfolded albumin induced by heat treatment at 80 °C and liposome membranes that enhanced sPLA2 accessibility to the bilayer membrane interface and caused hydrolysis of membrane phospholipids. In contrast, there was no difference between the albumin activity in the sera from 24-year-old subjects when treated either at 22 °C or at 80 °C (Figure 1 Columns 1 and 2), indicating that albumin in the serum of the young subject was more heat-stable than the serum albumin of 74-year-old subjects.

After pre-treatment at 80 °C, a mixture of equal amounts of serum from a 24-year-old subject and 74-year-old individual displayed markedly lower albumin activity (Figure 1 Column 5) than the effects of serum from a 74-year-old subject after 80 °C treatment (Figure 1 Column 4) (*p* < 0.0001). To determine that the inhibitory effect was not caused by the assays, a mixture of sera from two 74-year-old subjects showed no inhibitory effect after 80 °C treatment (Figure 1 Column 6). Because the Column 4 results were at the optimal level, the values shown in Column 6 were not the sum of two Column 4 results. Therefore, the values shown in Column 5 suggest the presence of a factor or factors in the sera of the 24-year-old subjects that may inhibit the conformational change of the 74-year-old subjects’ serum albumin at 80 °C, and the inhibitory moiety appears to be absent in the 74-year-old person’s serum. It is not clear why the combination of the 74-year-old and 24-year-old persons’ sera pre-treated at 80 °C (Figure 1 Column 5) yielded lower activity than the sera of 24-year-old and 74-year-old subjects alone (Figure 1 Columns 1–3). It is possible that sera from 74-year-old subjects can inhibit the albumin effect of the serum from a 24-year-old subject on the sPLA2 activity (Figure 1 Column 5 vs. Columns 1 and 2). In addition, we suspect that serum from young subjects may suppress the hydrophobic effect of misfolded albumin in the serum from elderly subjects (Figure 1 Columns 3 and 4).

The temperature effect on the albumin activity was best represented by the ratio of the albumin activity of serum sample treated at 80 °C divided by the albumin activity of the same serum sample treated at 22 °C (Figure 2). The ratio of the twelve serum samples from the 24-year-old individuals was 1.035 ± 0.088, and the ratio of the twelve serum samples from the 74-year-old subjects was 1.634 ± 0.065. The ratio of the 74-year-old person’s serum was markedly higher than the ratio of the 24-year-old person’s serum (*p* < 0.0001). The ratio of the albumin activity represents the reliability of the assays of sera-treated at 22 °C and 80 °C performed at the same time that could minimize the variable factors in the day-to-day assays, serum sample variations from person to person, and in reagent preparations.

## 3. Discussion

Human serum albumin in its pure form can undergo structural alterations under heat treatment up to 60–70 °C [16,17]. In this study, we demonstrated that albumin in the serum of 74-year-old subjects, after brief heat treatment of the serum at 80 °C, exhibited strong hydrophobic effects on bilayer membrane liposomes in the assay mixture. This indicates that the hydrophobic effect of albumin was due to exposure of albumin’s non-polar amino acids to the surface to interact with liposomal membrane phospholipids. In native albumin, the non-polar amino acids are buried in the core of the globular protein. Thus, the results suggest that albumin in the heat-treated serum of 74-year-old subjects was heat unstable and turned into a misfolded/denatured form. In contrast, the heat-induced albumin hydrophobic effect was not observed in the serum of 24-year-old individuals, suggesting a relatively stable protein in the young person’s serum. We previously observed that, without heat treatment, albumin in the sera of 74-year-old subjects also showed strong hydrophobic effects that could be determined by using high-membrane-fluidity liposomes [15]. Thus, we conclude that albumin in the serum of 74-year-old person is structurally different from the 24-year-old person’s and is likely in the misfolded form. Albumin misfolding in the sera of older persons may be, at least in part, a result of structural modifications due to age-dependent oxidative stress [18].

Although serum albumin misfolding itself may not be disease specific, apart from the cause of human serum albumin structural changes, the hydrophobic effect of misfolded albumin can alter membrane integrity of injured cells in the presence of the inflammatory response protein sPLA2, which may further exacerbate cell membrane damage. Alteration of membrane integrity may trigger harmful cell-derived signaling. Misfolded albumin may also lose its ability to carry out its normal homeostatic functions and may promote the misfolding of other protein/peptide and their aggregations in various disease states.

## 4. Materials and Methods

Sera were collected from 24-year-old young, healthy volunteers and 74-year-old healthy subjects, described previously [15]. Prior to the assay, an aliquot of 5 µL of each serum sample was mixed with 15 µL of water and the mixture was incubated either at 22 °C or at 80 °C for 10 min, followed by placing the mixture on ice. An aliquot of 4 µL of the pre-treated serum-water mixture containing 1 µL of serum was used to determine the albumin activity in a well of a 96-well microplate (Porvair PS White, PerkinElmer Inc., Waltham, MA, U.S.A.) by the method of the fluorescent low-membrane-fluidity liposome assay, detailed elsewhere [15]. Each serum sample was assayed in duplicates and the 22 °C-treated serum assay and the 80 °C-treated serum assay were conducted side by side at the same time.

## Figures and Tables

**Figure 1 ijms-23-11675-f001:**
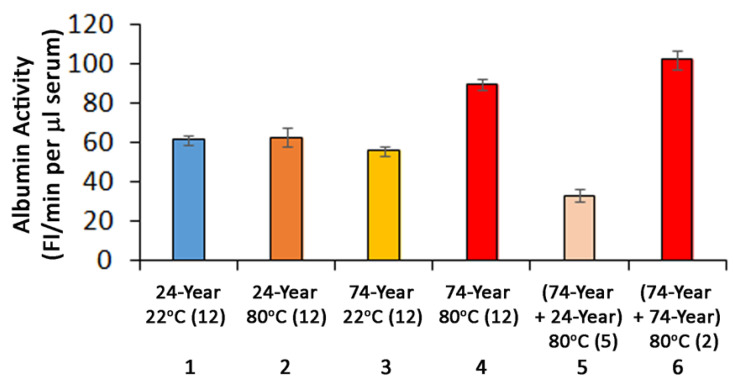
Determination of heat stability of serum albumin. The “Albumin Activity” represents the fluorescence intensity change resulted from albumin-membrane interactions, albumin-FA binding, and sPLA2 catalytic activity in the assay. Each serum sample was assayed in duplicates and the average was used for the albumin activity calculation. Each column represents the average of albumin activity in the sera of the number of subjects shown in the parenthesis. The bar represents mean ± SEM. The values of each column are: Column 1, 61.189 ± 2.525; Column 2, 62.510 ± 4.713; Column 3, 55.717 ± 2.254; Column 4, 89.881 ± 2.704; Column 5, 33.013 ± 3.470; and Column 6, 102.271 ± 4.833. The results of Columns were compared by Student’s *t*-test. A *p*-value < 0.05 was considered statistically significant. Similar results of Columns 1–4 in unit of (FI/min per µg serum albumin) were also observed (1.068 ± 0.042, 1.095 ± 0.088, 1.110 ± 0.081, 1.778 ± 0.106, respectively). The albumin level of each serum sample was previously determined [15].

**Figure 2 ijms-23-11675-f002:**
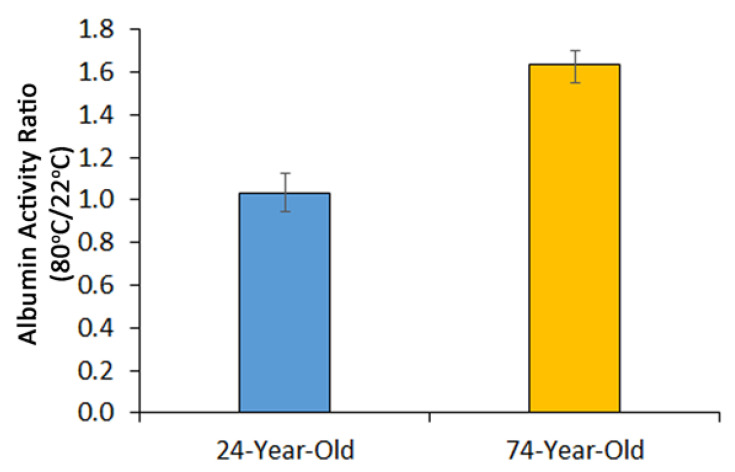
Albumin activity ratio. From the results of Figure 1, the albumin activity of serum sample treated at 80 °C was divided by the albumin activity of the same serum sample treated at 22 °C. The bar represents mean ± SEM of 12 samples, *p* < 0.0001. The value of Column 24-Year-Old is 1.035 ± 0.088, and the value of Column 74-Year-Old is 1.634 ± 0.065.

## Data Availability

Data were generated during the study.

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
