# Peer review of "Human Serum Albumin Misfolding in Aging and Disease"

_ijms, 2022, doi:10.3390/ijms231911675_

Round 1

Reviewer 1 Report

I read the manuscript "Human Serum Albumin Misfolding in Aging and Disease". Overall, I don't think this paper meets the criteria for publication in the Journal IJMS. I fail to see the impact of this manuscript. 

This communication looks like a review of the authors' recent published papers (the bibliography only counts 6 citations, 5 of which are from the authors), no additional background was described to highlight the novelty and the impact of the publication. Communications may not require the same standards as full articles, but this manuscript needs extensive editing, starting from the background/literature references. 

Looking at the results, the graph reported is from a single experiment. The statistical analysis is described, but not plotted in the graph. Why did authors choose to precisely analyse 74 and 24-yo individuals? It seems a highly restrictive selection, I would suggest to enlarge the sample diversity, as well as the sample size to have a more realistic variation.  

Apart from this, changes are needed in the English, as several paragraphs have long sentences that result in poor clarity.  

Reviewer 2 Report

I have some comments on the manuscript entitled “Human Serum Albumin Misfolding in Aging and Disease”.

1.       Write limitations of your study at the end of the abstract section.

2.       Significant value is missing in the histograms.

3.       Elaborate the communication section a little bit.

4.       Complete editorial checking will be needed to correct the grammatical and punctuation mistakes.

Round 2

Reviewer 1 Report

I see and appreciate the extensive work done by authors to improve their manuscript, but I'm not changing my mind. More and more relevant results are required to publish a communication in ijms, so once again i support rejection of this ms.